# Hydration Dynamics and the Future of Small-Amplitude AFM Imaging in Air

**DOI:** 10.3390/molecules26237083

**Published:** 2021-11-23

**Authors:** Sergio Santos, Tuza A. Olukan, Chia-Yun Lai, Matteo Chiesa

**Affiliations:** 1Department of Physics and Technology, UiT The Arctic University of Norway, 9037 Tromsø, Norway; tuza.a.olukan@uit.no (T.A.O.); chiayunlai@gmail.com (C.-Y.L.); matteo.chiesa@uit.no (M.C.); 2Laboratory for Energy and NanoScience, Masdar Institute Campus, Khalifa University of Science and Technology, Abu Dhabi 127788, United Arab Emirates

**Keywords:** AFM, small amplitude, hydration, water, multifrequency

## Abstract

Here, we discuss the effects that the dynamics of the hydration layer and other variables, such as the tip radius, have on the availability of imaging regimes in dynamic AFM—including multifrequency AFM. Since small amplitudes are required for high-resolution imaging, we focus on these cases. It is possible to fully immerse a sharp tip under the hydration layer and image with amplitudes similar to or smaller than the height of the hydration layer, i.e., ~1 nm. When mica or HOPG surfaces are only cleaved, molecules adhere to their surfaces, and reaching a thermodynamically stable state for imaging might take hours. During these first hours, different possibilities for imaging emerge and change, implying that these conditions must be considered and reported when imaging.

## 1. Introduction

The atomic force microscope can be operated in air, ultra-high vacuum (UHV), and liquid environments [1]. Nevertheless, a multiplicity of phenomena controlling cantilever dynamics [2], modes of operation [3], and feedback mechanisms [4,5] influence the possibilities and challenges encountered in each environment. This calls for attention in terms of the operation, utilization, and interpretation of data; for this reason, different theories have been developed and implemented in many cases [6]. The specific properties and morphology of samples [7], such as elasticity or viscoelasticity, [8] must also be considered when operating the instrument. Moreover, the user must cope with and understand the character of some dynamic variables, such as the sharpness of the tip [9,10], otherwise referred to as the tip radius, R, in high-resolution imaging applications. Perhaps counter-intuitively to the newcomer, the field has rapidly advanced in two extremes—in liquid [11,12,13,14] and UHV environments [15]—while several complex phenomena have hindered the imaging and quantification of phenomena in air [16] with similar resolutions, controls, or throughputs [3]. There has been research in air in terms of capillary interactions [17], spontaneous capillary condensation [16], and the way the air environment affects surfaces [18], molecules on it, and modes of imaging [3,19,20,21]. In terms of force measurements in air, most have focused on capillary interactions [22] arising from the formation and rupture of a capillary neck as a function of the relative humidity (RH), either experimentally [16] or theoretically [23]. Here, a capillary neck between the tip and the substrate forms and ruptures. In dynamic AFM, it is argued that the neck forms and ruptures [22,24] during each oscillation cycle, implying that the process is stable enough to occur at a rate of thousands of times per second. Some models consider the constant volume approximation [25], while others argue that the condensation from water in the neighborhood of the tip sample junction is also responsible for the volume of the neck [16]. The constant volume approximation assumes that the water layers on the surfaces contribute to the neck. Some have produced models to predict the values of formation and rupture of the neck at distances d_on_ and d_off_, respectively [22].

Force reconstruction techniques in air are currently being employed to decouple oscillatory and monotonic terms and simultaneously resolve interfacial water structures in liquid with an atomic resolution [12]. The oscillatory contributions correspond to the surface forces that provide information about the spatial frequencies of the liquid density. The monotonic term, many times neglected by the community developing the 3D mapping force technique [16], is the component of the force typically studied by force reconstruction methods that rely on averaging. It is worth noting, in the context of the topic under consideration, that even current research tends to focus on [26] the distinction between dry conditions and high-humidity conditions, i.e., extremes, and might obviate the aging process of the surface [27]. In short, while extremes, i.e., dry conditions and high relative humidity (RH) conditions, are sometimes considered by the community, in 2018, some [18] emphasized that the dynamic processes occurring on the surfaces due to nanoscale contamination are still largely neglected. This is even the case when the state of the surface is known to affect properties such as conductance [28], and research on surface contaminants and the quantification of interfacial properties in air is ongoing [29]. Arguably, less consideration has been paid to the phenomena arising from hydration dynamics and surface contamination as surfaces age [30] in relation to the alternatives that emerge for imaging. That is, adsorption kinetics is a process that occurs in time, and this can result in the emergence or absence of different imaging modes [21,31] also in time. Here, we explicitly interpret the problems that can arise when imaging in air due to the dynamics or adsorption kinetics of water molecules and other surface adherents. Previously, we have discussed the different regimes that are available when imaging in air with small amplitudes [32,33] in two imaging extremes, i.e., fresh and ambient-aged samples in air environments. We have also discussed the variations of force measurements as a function of time [27,34] and the effects on AFM measurements due to the presence of water films on surfaces [35]. The latter line of research provides information about the evolution of force profiles, but the connection between such evolution and the emergence of force regimes is stated mostly, or only, with respect to extreme conditions, i.e., fresh samples and aged samples.

In this work, we emphasize that the connection between the understanding of hydration dynamics, or adsorption kinetics, of contaminants such as water gained from force measurements and the possibilities for imaging is still largely lacking. Since these two objects of investigation are connected, it should be possible to understand how and when they affect each other. Thus, in this brief article, we discuss possible ways to connect the analysis of force reconstruction in time and as surfaces age on the one hand, and the evolution of imaging regimes in air and the possibilities that these offer on the other.

## 2. Discussion

### 2.1. The Problem of Small Amplitude Imaging and Surface Aging

The recent work by Eichhorn and Dietz [3] is particularly suitable to investigate the problems that might emerge when (1) imaging in air with small or ultra-small amplitudes and (2) advancing the field of multifrequency AFM, where several cantilever modes are excited in multiple ways, i.e., flexural and torsional. The first point is relevant because (1) such small amplitudes are of the same order of magnitude or smaller than the water films that form on surfaces, i.e., ~0.1–1 nm, and (2) it shows that even recent research fails to report the actual regime of operation, i.e., how many regimes were available for imaging in the experiments and which regime was exploited. This may be due to the fact that research in this field is still lacking. The second point is relevant because the work is recent, state-of-the-art, and possibly indicates how high-resolution and high-throughput multifrequency methods can or should advance in the future.

The problem can be stated as follows: in their approach, the authors used ultra-small oscillation amplitudes in order to reach sufficiently small tip–sample distances and thus increase resolution [19]. Nevertheless, since several amplitude branches might be available depending on the condition and state of the hydration layers, the actual imaging regime that was exploited remains ambiguous. Here, reproducibility also becomes a problem, for two main reasons. First, different imaging regimes might require varying the operational parameters depending on the conditions of the surface in order to be accessed. Second, other groups might fail to reproduce the results if the imaging regimes available and the conditions of the surface are not reported together with the operational parameters that have been used. Furthermore, what small amplitude means remains arbitrary without considering cantilever, sample, and environmental parameters since the limit to stability with small amplitudes prescribes what small means [20,36]. Next, different possibilities are discussed with a view to exploiting imaging regimes in both standard dynamic AFM and multifrequency AFM in air while considering such phenomena.

### 2.2. Force Reconstruction and Imaging

Previous reconstruction experiments, as a function of surface aging, show that variations in the tip–surface force can be dramatic, even for a given tip and sample as the surfaces age in air (Figure 1). The phenomenon that we term “aging” in this work relates to the changes that occur on a surface upon exposure to the air environment [35], including the effects that varying temperature and relative humidity (RH) might have on it over time [16]. The processes and phenomena can be molecularly identified with airborne molecules, such as hydrocarbons or water molecules [37,38], adhering to surfaces over time. The result is the formation of an interface between the air and the “pure” surface that leads to a modification of the effective surface properties [12,39]. Molecules might permanently or semi-permanently adhere [39] and there may or may not be order and structure in such nanometric films, depending on the surface’s atomic structure and composition [16]—even when the surface is immersed in water [12]. Over the past decade, our group has been mostly interested in the role of water adsorption and the resulting adsorption kinetics and corresponding variations on nanoscale forces [27,34,35,40]. Nanoscale forces can be investigated by means of force–distance profiles, force of adhesion maps, or other methods [33]. In particular, in Amplitude Modulation AFM (AM AFM), a dynamic mode of AFM operation, the standard observables, i.e., amplitude A, phase shift ϕ, and cantilever separation z_c_, can be turned into force distance curves (Figure 1a,b) by exploiting transformation algorithms [41,42,43,44,45]. The parameters that contain information about the cantilever and the environment, i.e., spring constant k, quality factor Q, and resonant frequency f_r_, must be known. There are procedures to calibrate and exploit these parameters when employing force reconstruction methods [46]. The forces that the tip and surface exert on each other due to surface forces are typically termed tip–sample force F_ts_, or simply F as in Figure 1, versus the distance d force or curve. In Figure 1b, the force has been normalised and termed F* by dividing the F by the absolute of the adhesion force F_AD_. The adhesion force F_AD_ is simply the minimum value of the force. All the data in Figure 1 are experimental data.

Figure 1a illustrates how forces vary as samples, such as mica or graphite, are exposed to ambient air. The sample in Figure 1a is highly oriented pyrolytic graphite (HOPG) and the sample in Figure 1b–d is graphene/Cu. The details on sample preparation for the graphene/Cu sample can be found elsewhere [47]. The surfaces of mica and graphite are crystalline surfaces that have the advantage of being easily cleaved. In this way, these surfaces can be cleaved and immediately imaged, i.e., within minutes, or be exposed to ambient air or controlled RH (see Ref. [40]). The graphene/Cu samples can be imaged by first annealing at high temperatures under UHV conditions and then exposing the samples to controlled RH. The samples can be re-annealed after the experiments to reproduce the findings. The main feature to be observed in Figure 1a is a “plateau” that forms, i.e., a range of distances where the force reaches minima and is relatively independent of distance, as a function of time. In order to investigate the evolution of this plateau, the force can be parametrized (Figure 1b) by computing the distance of force of adhesion ΔdF_AD_ and the area under the curve A_AD_. For more details, refer to the interpretation of these metrics in the literature [34,40]. The results obtained for the graphene/Cu sample (Figure 1c,d) show a similar behaviour to that of Figure 1a (graphite). In Figure 1c,d, the mean values and standard deviations are reported. Both parameters increase in the first hour after exposure to high RH and then drop after further annealing.

Clearly, the variations in force observed for the graphite sample in Figure 1a and for the graphene/Cu sample in Figure 1c,d induce a variation in cantilever dynamics for a set of operational parameters that also occurs in time. The implication is that the cantilever will behave very differently, and the surface will be probed very differently, in the first minutes after cleaving the sample in relation to hours later. A practical result for imaging is that different regimes of operation should become available with time. In our experience, after several hours of being exposed to the environment, the behaviour stabilizes. For this reason, the user must consider these variations and not expect that a given sample and setup will lead to certain results, irrespective of the dynamics of the “aging” process of the surface. In our work, we have only investigated the regimes of operation in two extreme regimes, i.e., freshly cleaved and aged after many hours, i.e., ~24 h or more. An example is discussed with the help of Figure 2.

In Figure 2, two extremes are shown for a mica sample. In Figure 2a,b, the amplitude A (dashed black lines) and the minimum distance of approach d_m_ (continuous black lines) are shown for an aged (Figure 2a), i.e., ~24 h, and a cleaved sample (Figure 2b), i.e., ~0–1 h. The minimum distance of approach per oscillation cycle can be obtained with the simple transform:(1)$$d_{m}=z_{c}-A$$
where both the cantilever surface separation z_c_ and A are known. Furthermore, while z_c_ is a relative value, since the absolute distance to the surface is not known experimentally, the difference with A is not. Several extra features manifest in both signals in the aged sample. It is important to note that for sufficiently large values of z_c_ or d_m_ A is quasi-constant and it is identified with what is typically termed free amplitude or A_0_. When A = A_0_, the tip–sample interaction is negligible. In terms of the behaviour in Figure 2a,b, A initially decreases with z_c_, and d_m_ is relatively independent of z_c_—with A decreasing in both cases. This oscillation branch is typically termed an attractive, low-amplitude [48] branch or L-state. For the cleaved surface, only this branch is available when the free amplitude is small enough [31,49]. Nevertheless, for sufficiently small values of A, there is a region of negative slope (Figure 2a) in A and positive slope in d_m_ [21]. This oscillation branch is not accessible for imaging in AM AFM, but has the advantage of driving the tip closer to the surface for small oscillation amplitudes, precisely because of the accentuated positive slope in d_m_. According to our simulations [32,49], the tip penetrates the hydration layer until perpetual water contact is made, i.e., the tip is in perpetual contact and immersed in the hydration layer. At this point, the amplitude starts to decrease with decreasing z_c_ (see peak in Figure 2a indicated by an arrow). This other branch is termed the SASS (Small Amplitude Small Set-point) branch [21]. However, it is important to realize that this oscillation branch is not accessible by simply setting a small amplitude. Rather, the hydration layer must be present, and the tip must go past the region of the negative slope. The first condition relies on the dynamics of hydration, and the second on operating the instrument with the understanding that several oscillation branches are available for a given set of operational parameters. In the SASS region, d_m_ decreases with decreasing A, implying that the smaller A is, the less the tip indents the sample (Figure 2a). Thus, for high-resolution and minimally invasive imaging, A or A_sp_, where sp stands for the set point when imaging, must be set as small as possible. Nevertheless, experimentally, the implication is that A_0_ cannot be made arbitrarily small. Rather, in both AM and Frequency Modulation FM AFM, the drive amplitude must be increased in order to stabilize the tip. This is because A_0_ controls the energy input into the system; therefore, the stability of the dynamics is against the non-linear interaction and noise. The experimentally reconstructed force versus distance profile is shown for both cases in Figure 2c. For the aged case, the “plateau” is observed as discussed when considering Figure 1a. Figure 1d shows the behaviour of HOPG in terms of the absolute of the adhesion force │F_AD_│ and the area under the curve A_AD_. Here, │F_AD_│ decreases rapidly in one hour and then remains relatively constant, but A_AD_ keeps increasing up to 24 h after cleaving. Our interpretation is that molecular adsorption is taking place, even though it does not affect F_AD_. This example illustrates that F_AD_ alone is not enough to parametrize the evolution of the attractive force as a function of time. After the experiment, the HOPG sample was annealed. A relation in terms of F_AD_ and contact angle is shown in Figure 2e for two cases; (1) when immediately probed after the cleaving case, (2) the case of being exposed to high RH for 24 h, and (3) after annealing. The mean values and standard deviations for F_AD_ are given for 15–30 data points. More details can be found in Ref. [40]. The procedure to obtain the contact angle is also described there, but a detailed explanation is outside the scope of this work.

Figure 3 is an illustration of the difference between imaging in (1) the attractive regime where the amplitudes A_sp_ are comparable to A_0_ (Figure 3a), (2) the SASS regime where A_sp_/A_0_ << 1 (Figure 3b), and (3) the attractive regime where the condition A_sp_/A_0_ << 1 also holds (Figure 3c). The inset in Figure 3d illustrates that the tip is fully immersed in the hydration layer when imaging in the SASS regime. The figure illustrates that the condition A_sp_/A_0_ << 1 for a given A_0_ can lead to different imaging regimes.

The reader can refer to the literature for a full description of the simulations describing the results for an amplitude A versus separation z_c_ curve when inputting the forces into the simulations, such as those in Figure 1a and Figure 2c for the aged case [49,50]. The simulation data allow to visualize a branch that is not observed in experimental Figure 2a, i.e., the repulsive regime or H-state. Only where A_0_ is sufficiently large [51] can this branch be experimentally reached [2].

### 2.3. Other Possible Dynamic Parameters to Consider

The definition of small amplitude is ambiguous because it depends on the cantilever, sample, and environmental parameters, i.e., when many parameters impose a limit on stability. One parameter that determines this limit is the tip radius R. Amongst others, the tip radius R controls the adhesion force F_AD_ since F_AD_ ≡ 4πγ, where γ is the surface energy [39]. The larger the attractive forces, the stronger the pull-off force and the more energy is required for the system to reach the surface. In dynamic AFM, this means that larger values of A_0_ are needed to reach the surface and the SASS imaging mode as R increases. For this reason, R must be considered when defining what small amplitudes mean. The spring constant k and the Q factor might also play a role [52]. Since the tip radius R can vary during experiments, the user must consider the state of the tip together with the state of the hydration layer in order to image with small amplitudes. We have carried out some experiments [49] to understand the consequences of attempting to reach the SASS regime with a relatively blunt tip, i.e., R > 20 nm. Nevertheless, these experiments were not systematic. It is likely that different force regimes are available for imaging, as surfaces also age as a function of R. This would be consistent with the fact that when R is of the order of magnitude of the water film, the Kelvin equation and other macroscopic behaviours might not hold [16,26]. Finally, it is worth noting that in the SASS regime, the mean deflection z_0_ can be employed to map the adhesion force F_AD_ while acquiring the standard topography. This is because the tip oscillates inside the hydration layer where the force is almost constant and approximately F_AD_ [33]. The expression to map the adhesion force becomes a simple transformation of the mean deflection map:(2)$$F_{\mathrm{AD}}={\mathrm{kz}}_{0}$$

Incidentally, and practically, monitoring a step-in deflection could be used experimentally to discriminate between the attractive regime and the SASS regime. It is likely, however, that as a systematic investigation of force reconstruction, aging, and imaging regime is carried out, new identities or useful expressions will emerge. Finally, the SASS regime can also be accessed and exploited in multifrequency AFM [3,53]. In this regime, sample indentation δ is minimal but non-zero. It can also be shown that the small amplitudes, when imaging in SASS, can be exploited to produce expressions to quantify surface properties [33].

## 3. Conclusions

In this work, we have reviewed and discussed (1) force measurements in air as a function of time and (2) small-amplitude imaging in dynamic AFM in the presence and absence of water layers. The objective has been to connect and coordinate theoretical and experimental work from these two topics to establish the key problems that might emerge when imaging as a result of the multiple and variable force regimes available. On the one hand, force measurements on freshly created surfaces show that the force presents an attractive and a repulsive component, qualitatively similar in shape to the standard Lennard Jones [54] profile. On the other hand, as surfaces age and as they are exposed to airborne water molecules, water films emerge on surfaces and force profiles develop a “plateau” of constant force that ranges from 1 to 2 nm, and that approximates the adhesion or minima in force. The work that we have carried out over the past decade has been discussed to show that this process is relevant over at least a few hours, depending on the surface, after the surface is created. Previously, we had shown that small-amplitude imaging, i.e., ~0.1–1 nm, for “aged” surfaces can lead to imaging in perpetual contact with the water layer while increasing resolution. Nevertheless, the possibility of imaging in conditions where (1) there are no water layers or (2) there are water layers is simplistic and binary. Rather, in agreement with the evolution of forces, the imaging regimes, and the way to reach them, should also be expected to be dynamic. The tip radius has also been shown to be a key variable when considering the dynamics of force regimes. In short, as AFM methods advance, it will become paramount to report not only operational, cantilever, and sample parameters, but also the state of the surface while imaging. The final point is that future research would benefit from coordinating the two lines of research that we have explored, i.e., force reconstruction and the emergence of imaging regimes as surfaces age.

## Figures and Tables

**Figure 1 molecules-26-07083-f001:**
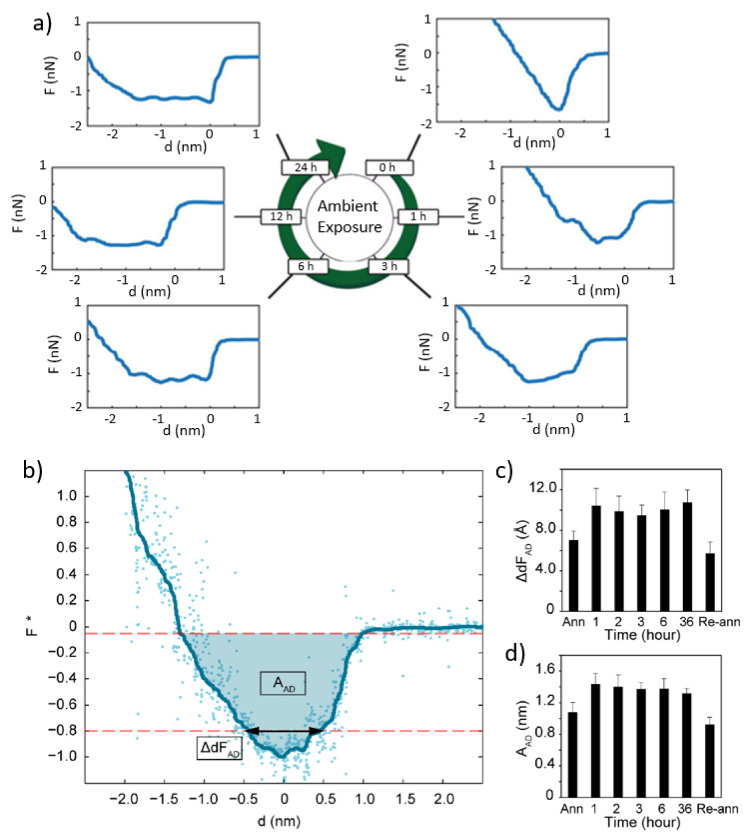
(**a**) Force versus distance curves obtained on a graphite sample as a function of time. The data have been acquired after cleaving the graphite (0 h, and 1 h, 3 h, 6 h, 12 h and 24 h) by keeping the samples inside a controlled environment in between experiments (the temperature was relatively constant within the 23 ± 2 °C interval and the relative humidity when acquiring the data were RH ~55 ± 5%). (**b**) Illustration of a normalized force versus distance curve obtained for a graphene/Cu sample and highlighting the meaning of the parametrization of the curve via ΔdF_AD_ and the area under the curve A_AD_. Mean values and standard deviations of (**c**) ΔdF_AD_ and (**d**) A_AD_ for the graphene/Cu sample. (**a**) reprinted with permission from reference [34]. Copyright 2018 PCCP Owner Societies. (**b**–**d**) reprinted with permission from reference [47]. Copyright, 2014 ACS Ltd.

**Figure 2 molecules-26-07083-f002:**
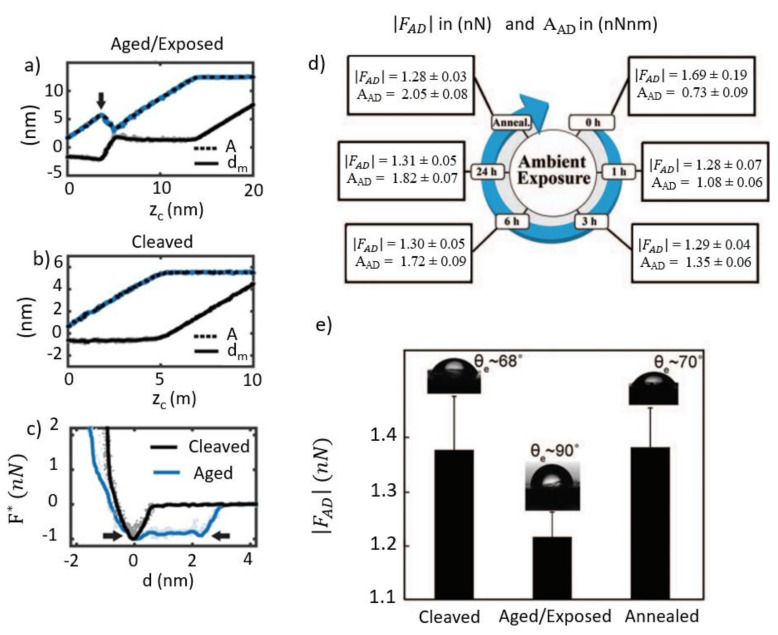
Amplitude A, separation z_c,_ and d_m_ data for the (**a**) aged and (**b**) cleaved conditions for a mica sample. (**c**) F* versus distance data, where the asterisk implies that the force has been normalized by dividing by the absolute of F_AD_. This has been performed for convenience in order to visualise the difference in behaviour with d for both the cleaved (black lines) and the aged (blue lines) conditions. (**d**) Evolution of |F_AD_| and A_AD_ with time of exposure in ambient conditions for an HOPG sample. (**e**) Evolution of the contact angle and |F_AD_| in three different cases for the HOPG sample. (**a**–**c**) reprinted, with permission from reference [31]. Copyright 2017 PCCP Owner Societies. (**d**,**e**) reprinted with permission from reference [40]. Copyright 2014 AIP Publishing.

**Figure 3 molecules-26-07083-f003:**
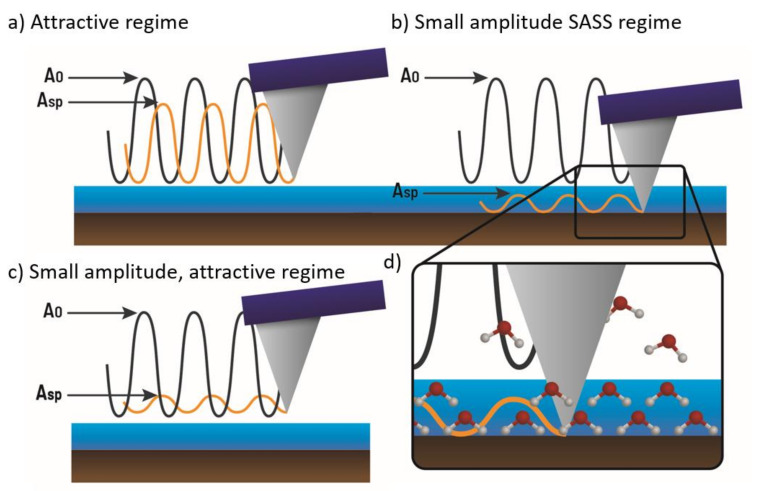
Illustration of a cantilever oscillating (**a**) above the hydration layer with amplitudes (A_sp_) comparable to A_0_ (standard attractive regime imaging), (**b**) with amplitudes A smaller than A_0_ when the tip is permanently immersed in the hydration layer (SASS regime), (**c**) above the hydration layer with small amplitudes (A_sp_) much smaller than A_0_ (Small amplitude attractive imaging). The inset in (**d**) is provided to illustrate that in the SASS regime the tip is immersed under the water layer where there might be structure and order. Airborne water molecules are also depicted for illustrative purposes.

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
