# Peer review of "Hydration Dynamics and the Future of Small-Amplitude AFM Imaging in Air"

_molecules, 2021, doi:10.3390/molecules26237083_

Round 1
Reviewer 1 Report
Santos et al. discuss the effect of hydration layers at ambient conditions in air on the imaging modes in dynamic AFM. The work highlights and addresses an important aspect that is very often neglected or just not considered, that implies the change of forces during time due to molecule adsorption with time. Parameters that influence the accessibility of the different regimes are also discussed.
The manuscript is well structured and nicely written. The content is needed to assure that scientist become clearer about important/possible imaging regimes in dynamic AFM and how to use them. The review fits well in the scope of the Journal. I can highly recommend publishing the manuscript with no or minor revisions. Minor suggestions for improvements are listed below.
- The authors might rethink their title. The effect of hydration dynamics is just one aspect they illuminate in their review.
- Could the authors comment on the influence of airborne molecules and the implications for the operation regimes?
- The sentence (page 2, line 93) “Typically, in our experiments, the samples are cleaved and directly imaged or placed in an enclosed petri dish with water and exposed to the air inside it.” should be rephrased. It is not exactly clear to me how they set up their experiment (placed in an enclosed petri dish with water and exposed to the air inside it?).
- Figure Caption, Figure 1: In line 102, the second “after” should be deleted.
- Is there an oscillation identifiable, corresponding to an oscillatory force originated by the hydration layer in the force curves of Figure 1 (plateau of minimum), similar to that in Ref. 16? If yes, could the periodicity be related with the water molecules or adsorbates?
- At the end of page 3, for the amplitude ratios, the free amplitude should be indexed by 0, i. e. Asp/A0, to be consistent with Fig. 3.
- What is the exact difference between the H-state and SASS and why is the H-state experimentally or physically not accessible using small free amplitudes?
- Could the authors specify reference 40?
- In Fig. 4 the color legend of the individual basins of attraction seems to be missing.
Author Response
Contents
Reviewer 1 Comments and Suggestions for Authors. 1
General comments. 1
Suggestions. 2
Reviewer 2 Comments and Suggestions for Authors. 8
General Comments and Suggestions for Authors. 8
Suggestions. 8
We thank reviewers and editors for the comments. Please find the answers below.
Reviewer 1 Comments and Suggestions for Authors
General comments
- Santos et al. discuss the effect of hydration layers at ambient conditions in air on the imaging modes in dynamic AFM. The work highlights and addresses an important aspect that is very often neglected or just not considered, that implies the change of forces during time due to molecule adsorption with time. Parameters that influence the accessibility of the different regimes are also discussed.
We are very satisfied with the summary of the reviewer. In particular we note that what the reviewer emphasizes is the highlight of our paper. Negatively, this is not a “results” paper since we are not producing new results. Positively, we are discussing forces and the effect of variations in forces in time due to surface adsorption in relation to the possibilities that these open for imaging. That is, we discuss our previous work on the evolution of force profiles in air environments in coordination with the several regimes that emerge for imaging via the variations in these force profiles. We also put that work into context with new insights brought about by sophisticated multifrequency approaches (see next question below). In short, the reviewer is correct that our paper is partly a review/perspective article.
- The manuscript is well structured and nicely written. The content is needed to assure that scientist become clearer about important/possible imaging regimes in dynamic AFM and how to use them. The review fits well in the scope of the Journal. I can highly recommend publishing the manuscript with no or minor revisions.
We thank the reviewer for noting this point. We believe that the main problem with this paper is 1) to first realize that we are discussing hydration dynamics to then 2) interpret how the hydration dynamics affect cantilever dynamics. So one thing is what happens to the surface and another thing is how what happens to the surface with time affects the resulting types of available tip-sample interactions for imaging.
Our understanding is that since the reviewer got our major point, it also made sense to the reviewer the type of figures and structure that we have thought about. Our aim was to bring to view and to coordinate our results and understanding in force reconstruction with our results when imaging. This is important, as the reviewer highlights and amongst other, because it will be a critical point for the development of the very sophisticated modes of imaging that are currently being developed under the generic name “multifrequency AFM”. In the paper we mention on of the latest and, in our opinion most complete, works in this topic, i.e., Eichhorn and Dietz(1), and base our discussion on these important results and the possible paths to advance in the field when considering them.
Suggestions
Minor suggestions for improvements are listed below.
- The authors might rethink their title. The effect of hydration dynamics is just one aspect they illuminate in their review.
Our title was “The effect of hydration dynamics when imaging in air in dynamic AFM”. We have now slightly modified the title to “Hydration dynamics and the future of small amplitude AFM imaging in air”.
Maybe this new title shows that we are talking about hydration dynamics more generally and not only about the effect when imaging in dynamic AFM. It also shows that we are talking about the future of small amplitude imaging in connection to hydration dynamics. Since the water films are of the order of nm it makes sense that small amplitude imaging will be particularly affected.
- Could the authors comment on the influence of airborne molecules and the implications for the operation regimes?
We thank the reviewer for addressing this point since it was our intention to interpret the evolution of force profiles with time, in relation to hydration dynamics, as a result of adsorption kinetics of water molecules, and possible hydrocarbons. That is, the adsorption of airborne molecules onto the surface modifies the surface forces and this, in turn, alters the available imaging regimes and the mechanisms for imaging. We have now emphasized this in the text.
- The sentence (page 2, line 93) “Typically, in our experiments, the samples are cleaved and directly imaged or placed in an enclosed petri dish with water and exposed to the air inside it.” should be rephrased. It is not exactly clear to me how they set up their experiment (placed in an enclosed petri dish with water and exposed to the air inside it?).
In agreement with the above and with reviewer 2 we have added a methods section where we now explain the process in detail. We have added
Sample preparation. We have been carrying out experiments on the aging of surfaces in air since 2013(2, 3) and have developed multiple ways to expose surfaces to airborne molecules, water vapour for the most, in a process that we have termed “aging”. Two extreme approaches consist of placing samples in a petri dish that contains 1) Milli-Q water or 2) silica gel. In the presence of Milli-Q water the relative humidity (RH) inside the petri dish, sealed with parafilm, is higher than 70%. In the presence of silica gel, where the petri dish is also sealed with parafilm, the RH is lower than 5%. When exposing the samples to Milli-Q water we term the process “aging” or “exposed” case. Here the water is deposited onto a petri dish filling it to approximately 50% of its volume. The samples are placed in the petri dish while avoiding contact between the sample’s surfaces and the Milli-Q water. Avoiding contact between the water can be achieved by placing a petri dish of smaller diameter inside the larger petri dish containing the water and placing the sample on top of the smaller petri dish. Using these procedures, we produce a temporal map of force distance curves such as that shown in Figure 1. More details regarding the procedure can be found elsewhere(3, 4). We obtained similar results in 2018(5) by exposing highly oriented pyrolytic graphite (HOPG) to a more controlled environment containing airborne water alone, i.e. water vapor. It can be shown that force curves obtained on HOPG samples can be recovered after exposure to high RH by annealing. In particular, the water layer can be removed by annealing at 150 C under vacuum conditions as discussed by Amadei et al.(6) for HOPG and by Lai et al.(4) for graphene on copper. The work by Lai et al. discusses the process of annealing in detail.
- Figure Caption, Figure 1: In line 102, the second “after” should be deleted.
The figure has now been extended and we have rewritten the caption. We have taken this into account. Thank you.
- Is there an oscillation identifiable, corresponding to an oscillatory force originated by the hydration layer in the force curves of Figure 1 (plateau of minimum), similar to that in Ref. 16? If yes, could the periodicity be related with the water molecules or adsorbates?
Sergio Santos’ work, together with Calò et al. (Ref 16), observed and reported some oscillations when recovering the force on CaF2 crystals. Nevertheless, we have never attempted to study oscillations and oscillations might be found on some surfaces depending on the structuring and stability of the layers on the surfaces. Being fair we believed this field is still to be explored. We could not report them, even if present, because we average the data over 15-30 force curves or more. That is, if there was any oscillation it would cancel out after averaging. To investigate the oscillations would require care when filtering and when averaging and considering possible drifting in the xy plane. In summary, we do not know and have not targeted such analysis, but we reported some similar behaviour in 2015.
- At the end of page 3, for the amplitude ratios, the free amplitude should be indexed by 0, i. e. Asp/A0, to be consistent with Fig. 3.
Thank you, we are writing A0 throughout now.
- What is the exact difference between the H-state and SASS and why is the H-state experimentally or physically not accessible using small free amplitudes?
We believe it is a matter of definition but also a matter of understanding the phenomena in detail. Since it is the objective of this work to explain these important details, we believe the reviewer does well to raise this question.
This point is interesting and has made us rethink the main writing. We hope the reviewer agrees with the changes we have made by considering this comment. First we will give an explanation here and then we will show how we have modified the manuscript accordingly.
In short, the terminology H-state and L-state comes from noting the emergence of two oscillation branches at a given cantilever-surface separation zc appearing as two distinct equilibrium states. This means that the cantilever will oscillate in equilibrium, that is, it can reach a steady state with constant amplitude at two different values of amplitude. The larger amplitude was termed high amplitude branch or high state of oscillation and the smaller amplitude low amplitude branch or low state of oscillation about 3 decades ago, when first reported, by Gleyzes et al.(7). They write for the first time
“In a frequency range just above the resonant frequency there exist stable oscillating states: A high amplitude state is observed while the driving frequency sweeps up; a low amplitude state is observed while the frequency sweeps down [Fig. 1 (a)]. A corresponding hysterisis is also observed for the phase of the oscillator, as shown in Fig. 1 (b).”
The terminology can also be found in a paper by Garcia and San Paulo (2000) where they claim the following(8):
The authors refer to the work by Gleyzes et al. and emphasize that they want to decipher the mechanisms by interpreting the numerical solution of the equation of motion. Otherwise, it is typical in dynamic AFM to speak of two regimes, i.e., the attractive regime where there is never mechanical contact with the surface during an oscillation cycle and the repulsive regime where there is intermittent mechanical contact. It is, in our opinion, standard that non experts in the field term the L state attractive regime and H state repulsive regime. Nevertheless, there are condition where the L and the H states coexist and conditions where only one or the other can be reached. Furthermore, it is not true that in the L state there is no intermittent mechanical contact. There might be depending on operational, cantilever and sample parameters. In the repulsive regime typically there is always intermittent mechanical contact. The interaction can also be understood as average attractive or average repulsive force, but all these are simplifications.
Importantly, we do not term the SASS regime H state, even though we believe that the SASS regime is a special condition of the emergence of the H state, because in the SASS regime there is perpetual contact with the water films on the surface. This phenomenon must be emphasized and for this reason we give it its own name. Furthermore, at separations zc where the SASS regime can be observed, there is only one solution to the equation of motion. Namely the SASS regime. The reviewer can see that the SASS regime is a continuation to the H state (Figure 4a) but there are these two differences mentioned above that make it a special region of interaction. Whether it is legitimate to claim that it is a different state or not might be debatable. Nevertheless, to our knowledge, Garcia et al and others carried out their simulations without paying particular attention to 1) these very small separations or 2) the fact that the presence of water films on the surfaces of the sample and the tip affect the cantilever dynamics. They did not consider these phenomena when solving the equation. We did consider these in our simulations, as the reviewer can find here(9) and in our latest review on water films on surfaces here(2).
- Could the authors specify reference 40?
Ref 40 is now 47.
Ref 40 is the PhD thesis of Sergio Santos. We have now added the title and where the thesis is published. At that time Sergio Santos had not developed methods to reconstruct the force in air. In particular, methods started to appear for AM AFM in air at around that time. We implemented our first working method to visualise the interaction from experimental data in 2013 and the first thing we used it for was to attempt to observe such phenomenon, i.e., the evolution of the force profile when aging surfaces in air.
- In Fig. 4 the colour legend of the individual basins of attraction seems to be missing.
Correct. It was also missing in the PhD thesis of Sergio Santos and appeared only in the caption. We have added a new version with colours in the legend.
Submission Date
07 October 2021
Date of this review
14 Oct 2021 15:13:27
Reviewer 2 Comments and Suggestions for Authors
General Comments and Suggestions for Authors
The draft is truly interesting and deserves to be considered for publication after major revision.
Suggestions
- The introduction should be extended, and authors should highlight previously published results regarding hydration dynamics(especially studies with force measurements).
We have included some extra references and a new paragraph in the introduction on force measurements. We have also referred the reader to classic work such as that by Butt et al. In particular we refer the reader to work on force measurements in relation to the capillary neck. As far as we know, there is not so much work on force in relation to the actual hydration layer in air apart from the latest work by Garcia’s group that we also reference there in terms of oscillatory forces due to periodicity and structure and order of the nanometric films that form on surfaces in air environments.
This is the paragraph
In terms of force measurements in air, most have focused on capillary interactions(10) arising from the formation and rupture of a capillary neck while imaging(11) or in simulations(12). Here, a capillary neck forms between the tip and the substrate. In dynamic AFM it is argue that the neck forms and ruptures(10, 13) during each oscillation in dAFM implying that the process is stable enough to occur at a rate of thousands of times per second. Some models consider the constant volume approximation (14) while others argue that condensation from water in the neighbourhood of the tip sample junction is also responsible for neck formation and rupture(11). Some have produced relatively simple methods to predict the values of formation and rupture of the neck at don and doff respectively(10). The topic of force measurements from a theoretical point of view is out of the scope of this work but a thorough review was produced by Butt and Kappl(15). Arguably however, less consideration has been paid to the phenomena arising from hydration dynamics as surfaces age(16) and the possible and different imaging modes(17, 18) that result from such phenomena.
- What is new and significant in the presented study from previously published should be additionally highlighted.
In the general comments to reviewer 1 (above) we indicate that the reviewer is correct that our work is not so much an article on new results. This is an article on a discussion and interpretation of how our past simulations and work on force measurements as a function of exposure of samples to ambient air, high and low relative humidity etc., can be interpreted in view to improve future methods for imaging with small amplitudes. In particular this is meant to be an invited article were we were asked to write about how our work on AFM and water films on the nanoscale can be employed to interpret and highlight the mechanisms for future work on high resolution imaging. Additionally, this work should discuss what we can learn about the hydration layer and highlight the effects of the hydration layer on imaging with small amplitudes. This is why we base our criticisms, positive and negative, on the most recent and relevant work in the literature on this topic. In our opinion that of Eichhorn and Dietz(1) is a very good starting point to advance the field from 2022 and we are focusing on mechanisms related to their results and our past work.
For example,
1) we discuss the different regimes that are available for small amplitude imaging in air
2) we discuss the importance of the hydration dynamics, i.e., the temporal evolution of the nanometric films that form on the surfaces with time via adherence of airborne contaminants like water molecules or hydrocarbons, when imaging.
We have dedicated almost a decade to investigate the first point above. We have also discussed the relevance that the nanometric films that form on surfaces have for imaging but
3) here we implicitly emphasize and discuss that the regimes of interaction emerge in time as molecules adhere to the surface. The implication is that the “method” of imaging, in terms of operational parameters and possibilities for imaging, depends on the dynamics or adsorption kinetics of water molecules and other adherents to the surface.
This means that what we have discussed in our previous work cannot be reduced to either 1) imaging without water films where no hydration or nanometric films are present or 2) with water films where these are present. Rather, the very dynamics of the surface in terms of adsorption kinetics and eventual thermodynamic quasi equilibrium must be considered.
In this respect, we have extended our main figures (Figures 1 and 2) on force reconstruction as a function of time, and connected them to imaging regimes.
In order to consider the comments by the reviewer we now emphasise all this in the introduction and when discussing Figures 1 and 2. We also emphasize what is knew, what is old and possible future work in the conclusions.
Results should be presented separately(section results), and I would like to see the error bars and discuss errors in the presented measurements.
Figures 1 and 2 now have error bars presented as a mean value and standard deviation (see Figures 1 and 2 on force reconstruction). This section is now called discusion and results. We are following the same structure as Calò et al. in their molecules paper here (19).
- Introduction
- Results and discussion
- Materials and methods
- Conclusion
The methods to acquire and present the data are found in the methods section.
We hope that the reviewer finds these changes appropriate and sufficient.
The discussion part should be extended, especially the description and physics behind the model used by authors to explain results with more details about the physics of hydration and forces at the nanoscale level.
We have included the models in some detail. Nevertheless we believe that the changes requested by the reviewer have made the manuscript very lengthy. We wanted to focus on a message, i.e., to be careful with operational parameters and the behaviour of the surface forces with time in terms of the several imaging regimes that become available. Currently the paper is relatively different from what we hoped.
For this reason we also hope that the reviewer understands the scope of our work and that adding more would make it too heavy. We hope the reviewer and editors agree with us and thank the reviewer for the comments.
The conclusion part should be more focused and detailed(almost complete changing is needed)
We have modified the conclusions fully.
In the end, I advise authors to present fewer figures(to group figures in a few panel figures) in order to have a more coherent story.
We have maintained the number of figures but included what the reviewer considered to be most important, i.e., a discussion on errors and a presentation on errors in force reconstruction. These main figures are now included in the manuscript, and we believe they help to form a coherent story.
All necessary technical details (about cantilever, measurements) should be transferred to supporting info.
We have added a supplementary document where details on parameters, calibration etc. can be found.
Based on the criticism mentioned above, I recommend major and obligatory revision of the draft before reconsidering for publication.
Submission Date
07 October 2021
Date of this review
01 Nov 2021 12:22:36
- A. L. Eichhorn, C. Dietz, Simultaneous Deconvolution of In-Plane and Out-of-Plane Forces of HOPG at the Atomic Scale under Ambient Conditions by Multifrequency Atomic Force Microscopy. Advanced Materials Interfaces n/a, 2101288 (2021).
- S. Santos et al., Investigating the Ubiquitous Presence of Nanometric Water Films on Surfaces. The Journal of Physical Chemistry C, (2021).
- C. A. Amadei, T. C. Tang, M. Chiesa, S. Santos, The aging of a surface and the evolution of conservative and dissipative nanoscale interactions. The Journal of Chemical Physics 139, 084708 (2013).
- C.-Y. Lai et al., A Nanoscopic Approach to Studying Evolution in Graphene Wettability. Carbon 80, 784-792 (2014).
- M. Chiesa, C.-Y. Lai, Surface aging investigation by means of an AFM-based methodology and the evolution of conservative nanoscale interactions. Physical Chemistry Chemical Physics 20, 19664-19671 (2018).
- C. A. Amadei, C.-Y. Lai, D. Heskes, M. Chiesa, Time dependent wettability of graphite upon ambient exposure: The role of water adsorption. The Journal of chemical physics 141, 084709 (2014).
- P. Gleyzes, P. K. Kuo, A. C. Boccara, Bistable behavior of a vibrating tip near a solid surface. Applied Physics Letters 58, 2989-2991 (1991).
- R. Garcia, A. San Paulo, Dynamics of a vibrating tip near or in intermittent contact with a surface. Physical Review B 61, R13381-R13384 (2000).
- S. Santos, Enhanced sensitivity and contrast with bimodal atomic force microscopy with small and ultra-small amplitudes in ambient conditions. Applied physics letters 103, 231603 (2013).
- L. Zitzler, S. Herminghaus, F. Mugele, Capillary forces in tapping mode atomic force microscopy. Physical Review B 66, 155436 (2002).
- M. R. Uhlig, R. Garcia, In Situ Atomic-Scale Imaging of Interfacial Water under 3D Nanoscale Confinement. Nano Letters, (2021).
- S. Leroch, M. Wendland, Influence of Capillary Bridge Formation onto the Silica Nanoparticle Interaction Studied by Grand Canonical Monte Carlo Simulations. Langmuir 29, 12410-12420 (2013).
- E. Sahagún, P. García-Mochales, G. Sacha, J. J. Sáenz, Energy dissipation due to capillary interactions: Hydrophobicity maps in force microscopy. Physical review letters 98, 176106 (2007).
- V. V. Yaminsky, The hydrophobic force: the constant volume capillary approximation. Colloids and Surfaces A 159, 181–195 (1999).
- H. J. Butt, M. Kappl, Normal capillary forces. Adv Colloid Interface Sci 146, 48-60 (2009).
- J.-Y. Lu, C.-Y. Lai, I. Almansoori, M. Chiesa, The evolution in graphitic surface wettability with first-principles quantum simulations: the counterintuitive role of water. Physical Chemistry Chemical Physics 20, 22636-22644 (2018).
- M. Alshehhi, S. M. Alhassan, M. Chiesa, Dependence of surface aging on DNA topography investigated in attractive bimodal atomic force microscopy. Physical Chemistry Chemical Physics 19, 10231-10236 (2017).
- S. Santos et al., Stability, resolution, and ultra-low wear amplitude modulation atomic force microscopy of DNA: Small amplitude small set-point imaging. Applied Physics Letters 103, 063702-063705 (2013).
- A. Calò, A. Eleta-Lopez, T. Ondarçuhu, A. Verdaguer, A. M. Bittner, Nanoscale Wetting of Single Viruses. Molecules (Basel, Switzerland) 26, (2021).

Reviewer 2 Report
The draft is truly interesting and deserves to be considered for publication after major revision.
The introduction should be extended, and authors should highlight previously published results regarding hydration dynamics(especially studies with force measurements). What is new and significant in the presented study from previously published should be additionally highlighted.
Results should be presented separately(section results), and I would like to see the error bars and discuss errors in the presented measurements.
The discussion part should be extended, especially the description and physics behind the model used by authors to explain results with more details about the physics of hydration and forces at the nanoscale level
The conclusion part should be more focused and detailed(almost complete changing is needed)
In the end, I advise authors to present fewer figures(to group figures in a few panel figures) in order to have a more coherent story.
All necessary technical details (about cantilever, measurements) should be transferred to supporting info.
Based on the criticism mentioned above, I recommend major and obligatory revision of the draft before reconsidering for publication.
Author Response

(The authors gave the same response as above.)

Round 2
Reviewer 2 Report
The authors provide detailed and correct answers to all referee's questions regarding the science and the presentation style (article form), and the revised version is significantly improved from the first. Based on that fact, I recommend the acceptance of the revised draft in unchanged form.
Author Response
REVIEWER 2: The authors provide detailed and correct answers to all referee's questions regarding the science and the presentation style (article form), and the revised version is significantly improved from the first. Based on that fact, I recommend the acceptance of the revised draft in unchanged form.
AUTHORS: Thank you.